# Novel Synthesis of Dihydroisoxazoles by *p*-TsOH-Participated 1,3-Dipolar Cycloaddition of Dipolarophiles withα-Nitroketones

**DOI:** 10.3390/molecules28062565

**Published:** 2023-03-11

**Authors:** Caiyun Yang, Sirou Hu, Xinhui Pan, Ke Yang, Ke Zhang, Qingguang Liu, Xiaobing Xin, Jie Li, Jinhui Wang, Xiaoda Yang

**Affiliations:** 1Key Laboratory of Xinjiang Phytomedicine Resource and Utilisation, Ministry of Education, School of Pharmaceutical Sciences, Shihezi University, Shihezi 832002, China; 2Stake Key Laboratory of Natural and Biomimetic Drugs, Department of Chemical Biology, School of Pharmaceutical Sciences, Peking University, Beijing 100191, China

**Keywords:** α-nitroketones, alkenes, alkynes, 1,3-dipolar cycloaddition, *p*-TsOH, isoxazolines

## Abstract

This article reports in detail a method for the synthesis of 3-benzoxoxazoline by the reaction of alkenes (alkynes) and a variety of α-nitroketones in the presence of *p*-TsOH. The scope of alkenes is broad, including different alkenes and the alkyne. This reaction provides a convenient and efficient synthetic method of 3-benzoylisoxazolines.

## 1. Introduction

Heterocycles are important structural elements, which are present in natural products from all classes and in many biologically active synthetic compounds [1]. Heterocyclic compounds perform an important role in chemical industry, e.g., food fragrance and dyes. Amongst these, isoxazole and its derivatives represent a group of five-element heterocyclic compounds containing oxygen and nitrogen atoms of a valuable class [2]. Additionally, they have performed a vital role in the theoretical development of heterocyclic chemistry and are also extensively used in organic synthesis [3]. Isoxazoles have attracted an increasing research interest, and are widely used and studied in the modern drug discoveries as non-classical amide or ester bioisosteres, and potential pharmacophores endowed, and most isoxazoles have strong biological activity [4,5].

Isoxazolines are partially saturated analogs of isoxazoles as important intermediates for synthesis of varieties of fascinating organic molecules applicable to both basic organic synthesis and life sciences [6]. Isoxazolines can be converted into various synthetic units, such as hydroxy ketones [7], amino alcohols [8], β-hydroxynitrile [9], and masked aldols [10], and be used as synthetic equivalent of 1,3-dicarbonyl structure [11]. Isoxazolines can exhibit a variety of bioactivities, such as anti-inflammatory [12], anticancer [13], hypoglycemic [14], antibacterial [15], anti-HIV [16], anti-Alzheimer’s [17], antifungal [18], antimalarial [19], antioxidant [20], anti-tuberculosis [21], and antinociceptive [22] activities (Figure 1). Isoxazolines can also be good herbicides [23] and insecticides [24]. Therefore, the development of new methods for more efficient synthesis has been always an attractive task.

Based on the characteristics and wide application of isoxazolines derivatives, the research progress of isoxazolines derivatives have progressed rapidly in recent years, and a large number of synthetic methods for isoxazole derivatives are reported in the literature every year. These approaches can be summarized as the following four major types: (1) 1,3-dipolar cycloaddition between nitrile oxide and unsaturated hydrocarbon [25,26,27,28,29], (2) intramolecular addition cyclization reaction of unsaturated hydroxime [30,31,32,33], (3) condensation reactions of 1,3-dicarbonyl derivatives [34], and (4) cycloisomerization [35]. In the past decades, the 1,3-dipole cycloaddition reaction of alkenes with nitrile oxide is the most direct and extensive method for the construction of isoxazoline skeletons [36]. Nitrile oxides are usually derived from aldoximes and nitro compounds [37,38], but the common use of transition metal catalysts, such as Cu(I), Cu(II), and Ru(II), in the reaction makes the products residual metal and cytotoxic [39,40,41], which limits its application in biology and drug development (Figure 1). Therefore, the development of practical, simple, and cost-effective new methods for synthesis 2-oxazolines would complement current methods.

In the previous study, we first used the alkaline catalyst chloramine-T to catalyze the reaction of α-nitroketone and alkene to synthesize isoxazoline with a yield of 77% [42]. In the present work, we report a novel synthesis of dihydroisoxazoles by *p*-TsOH (anhydrous)-participated 1,3-dipolar cycloaddition of isoxazoline with *α*-nitroketones. On one hand, compared with the strong acid (H_2_SO_4_)-catalyzed synthesis of isoxazole [43], *p*-TsOH gives a milder reaction condition that avoids carbonization of organic substance, and it is low in toxicity and is inexpensive. On the other hand, Natarajan Arumugam and co-workers [44] reported a good, facile, and efficient method for the rapid synthesis of fused pyrrolidine and indolizinoindole heterocycles through 1,3-dipolar cycloaddition in the presence of *p*-TsOH. Additionally, Zhenghui Guan and co-workers [45] also demonstrated a *p*-TsOH mediated 1,3-dipolar cycloaddition approach of nitroolefins and sodium azide for the synthesis of 4-aryl-NH-1,2,3-triazoles, and a slightly higher yield (93%) was isolated. It is an efficient *p*-TsOH-mediated 1,3-dipole cycloaddition reaction that can tolerate a wide range of functional groups, and quickly and easily obtain the target product under mild conditions. *p*-TsOH was discovered as a vital additive in this type of 1,3-dipolar cycloaddition. Herein, isoxazolines, given the importance of preparing biologically active molecules, are chosen for validation of the accessibility, operational simplicity, and atom economy of our method.

## 2. Results and Discussion

### 2.1. Optimization of the Reaction Conditions

First, we compared the effects of different acids and solvents. The reaction of benzoylnitromethane **1a** with allylbenzene **2a** to form isoxazoline **3a** was performed and the results were summarized in Table 1. It is noted that the reaction without acid did not proceed. In the presence of various acids, i.e., HCl, HNO_3_, H_2_SO_4_, TFA, H_3_PO_4_, fluoroboric acid, MsOH, and *p*-TsOH, the yield was significantly improved. It appears that oxidative acids produced similar good yield (Table 1, entries 3 and 4), but reagents used there are expensive, toxic, and dangerous. It was found that MsOH in *i*-PrOH at 80 °C effectively promoted the formation desired isooxazoline. However, the yield of **3a** was slightly lower than that of *p*-TsOH (LD_50_: 1410 mg/kg) (Table 1, entries 8 and 9), and MsOH (LD_50_: 200 mg/kg) is highly toxic. Therefore, *p*-TsOH, which is non-oxidizing, corrosive, and low toxic was selected to participate in the synthesis of isoxazoline with a yield of 67%. While among the five tested solvents (ACN, *i*-PrOH, DMF, DMSO, and H_2_O), it was found that the yield reduced significantly with the relative polarity of the solvent (Table 1, entries 9–13) and can obviously is the best solvent.

The reaction temperature and amount of *p*-TsOH were further optimized. The results were shown in Figure 2. Then, the optimal condition was regarded acanACN as solvent, 4 equiv of *p*-TsOH was involved in the reaction at 80 °C for 22 h, in which a good yield of 90% could be obtained.

### 2.2. Substrate Scope Studies

With the optimal reaction conditions, we first tested the 1,3-dipolar addition reaction for benzoyl nitromethane and allylbenzene and had a yield of product of 90%. Then, to examine the generality and scopes of this methodology, we took a variety of benzoylnitromethane derivatives **1** (Figure 2) and allylbenzene **2a** as substrates and representative results were shown in Figure 2. These results showed that a variety of electronically varied aromatic α-nitroketones were well compatible with the cycloaddition in all the reactions, and reaction generally obtains in moderate to good yields for the synthesis of isoxazoline derivatives. Moreover, in this reaction, we found a good regioselectivity, which was consistent with the work of Ken-ichi Itoh [45].

At first, it was found that the R_1_ substituents would affect the cycloaddition efficiency in these reactions. The electron-rich α-nitroketones (**1e**, **1g**, Figure 2) provided products (**3e**, **3g**, Figure 2) in slightly better yields in comparison to the electron-deficient ones (**1d**, **1f**, **1i**, Figure 2). Different electron-withdrawing substituents at the same position of phenyl-α-nitroketone resulted in similar yields (**3d**, **3i**, Figure 2). Additionally, surprisingly, isoxazoline derivative (**3f**, Figure 2) were obtained in moderate yields when electron-donating group(**-OMe**) were used with a yield of 66%, which was close to the yield of isoxazoline obtained by electron-deficient α-nitroketone. In the case of electron-deficient α-nitroketone, the corresponding were obtained in good yields (**3b**–**3d, Figure 2**), respectively 73%, 70%, and 67%. The results show that the position of the substituent has no effect on the reaction results. Additionally, aromatic substrate, such as benzene, behaved similar to an electron-withdrawing substituents and gave a yield of 68% for product **3h**. However, the reaction rate was slower than that of aliphatic substituted substrates.

Next, we also investigated the scope of the alkenes (Table 2). The reaction of **1a** with alkenes derivatives **4** was carried out under the optimum reaction conditions whose results are shown in Table 2. All the reactions gave **5a**–**5f** as product, respectively, in good to excellent yield, except **5f**. The type of reaction substrate alkenes was modified, and it was found that the reaction proceeded well with both aliphatic alkenes and aromatic alkenes affording isoxazolines in good yields from the same *α*-nitroketone (entries 1–5, Table 2). In addition, cycloaddition of cyclohexene (**4f**) with benzoylnitromethane (**1a**) could also be achieved in fairly good yields, the corresponding isoxazoline **5f** was obtained in 69% yield (entry 6, **5f, Table 2**).

In addition, in order to expand the applicability of the reaction, we further examined the types of reaction materials. Isoxazolines were synthesized using the dipolarophiles **2a** (allylbenzene) and alkyl nitroketones **6** (Figure 3). The results showed that in the presence of *p*-TsOH, alkyl nitroketones were also able to react with dipolar reagents to obtain isoxazoline derivatives; unfortunately, compared with **3** and **5** phenylisoxazoline, **7a** and **7b** yields were lower, 23% and 20%, respectively.

Finally, the alkyen **8** was used for the reaction with α-nitroketone (**1a**) under the optimized reaction conditions, which obtained in excellent yields Isoxazoles. Under the same conditions, the reaction rate with alkyen was quicker than with alkenes. Nevertheless, **9a** and **9b** were obtained in 85% and 88% yield, respectively (Figure 4).

### 2.3. Mechanistic Studies

After screening the reaction conditions and studying the application of the products, the reaction mechanism was also studied. On the basis of the reaction mechanism reported by Ken-ichi Itoh et al. [46,47]. We proposed the theory of 1,3-dipolar cycloaddition of benzoylnitromethane with allylbenzene was deduced as follows (Figure 5): In this reaction, α-nitroketones are converted to nitroso cations in the presence of non-aqueous phase protons, then nitrile oxides are formed from nitroso cations. Finally, isoxazolines and their derivatives are obtained by intermolecular the 1,3-dipolar cycloaddition cyclization of dipolarophiles (alkenes or alkynes) and nitrile oxide.

## 3. Materials and Methods

### 3.1. General Experimental Methods

The structures of produced compounds were firmly confirmed by ^13^C NMR and ^1^HNMR spectra, and supported by HRMS, and IR data (see the Appendix A).

^1^H NMR (400 MHz) and ^13^C NMR (101 MHz) were recorded at room temperature on DRX-400 spectrometer (Bruker, Saarbrücken, Saarland, Germany) in CDCl_3_. The chemical shifts are given in parts per million (ppm) on the delta (δ) scale. The solvent peak was used as a reference value, for ^1^H NMR: CDCl_3_ δ_H_ 7.26; for ^13^C NMR: CDCl_3_ δ_C_ 77.16 ppm. IR spectra were recorded using an Avatar 360 FT-IR ESP spectrometer (Nicolet, Madison, Wisconsin, USA) at room temperature. HR-ESI-MS spectra were acquired using an Agilent 6210 ESI/TOF mass spectrometer (Agilent Technologies, Santa Clara, CA, USA). Analytical TLC was conducted on silica gel plates (GF254, Yantai Institute of Chemical Technology, Yantai, China). Spots on the plates were observed under UV light. Column chromatography was performed on silica gel (200~300 mesh and 300~400 mesh; Qingdao Marine Chemical Factory, Qingdao, China). Super-dry solvent *i*-PrOH, ACN, DMSO and DMF were purchased from Aldrich and used as supplied. The α-nitroketones were synthesized using the same method as reported in the literature [48,49].

### 3.2. General Procedure for the Cycloaddition of Alkenes and α-Nitroketones

*p*-TsOH (0.500 mmol, 4 equiv) was added to a solution of **1** (0.125 mmol, 1 equiv) or **6** (0.125 mmol, 1 equiv) and **2** (0.625 mmol, 5 equiv) (or **4** (0.625 mmol, 5 equiv) or **8** (0.625 mmol, 5 equiv)) in ACN (0.2 mL). The mixture was then stirred at 80 °C until the starting material disappeared as monitored by TLC. Subsequently, the mixture was directly purified by flash chromatography (with ethyl acetate/petroleum ether as the eluent) to obtain the desired product (**3**, **5, 7** or **9**).

## 4. Conclusions

In conclusion, we have developed an efficient cycloaddition of a variety of α-nitroketones with alkenes or alkyne using inexpensive and gentle acid. Among the synthesized compounds, the yield of cycloaddition products of substituted phenylnitroketone is high (66–90 %), while the yield of cycloaddition products of alkyl nitroketones (such as **7a**–**b**) is low (20–23 %). This synthesis is based on cycloaddition 1,3-dipolar in presence of *p*-TsOH, which is attractive that the low cost, simple synthetic route, and ease of handling of the gentle acid. It is particularly noteworthy that the reaction provides an effective synthesis method for 3-carbonylisoxazolines. The development of other methods for the synthesis of 3-carbonylisoxazoline is currently under investigation and will be disclosed in due course.

## Data Availability

The data presented in this study are available in the article and Appendix A.

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
