# Peer review of "Novel Synthesis of Dihydroisoxazoles by p-TsOH-Participated 1,3-Dipolar Cycloaddition of Dipolarophiles withα-Nitroketones"

_molecules, 2023, doi:10.3390/molecules28062565_

Round 1

Reviewer 1 Report

The manuscript by Yang et al. describes the use of pTsOH for the synthesis of dihydroisoxazoles. The work is interesting. The abstract briefly describes the work done, however, yield range of the synthesized compounds should be mentioned in the abstract. The introduction is well written and relevant examples from the literature have been added. The authors have applied their methodology on the already reported molecules. I suggest authors should also provide some new molecules reflecting the applicability of their methodology in addition to already reported molecules.

Author Response

The work is interesting. The abstract briefly describes the work done, however, yield range of the synthesized compounds should be mentioned in the abstract. The introduction is well written and relevant examples from the literature have been added. The authors have applied their methodology on the already reported molecules. I suggest authors should also provide some new molecules reflecting the applicability of their methodology in addition to already reported molecules.

Response: Thank you very much for pointing this out. This manuscript is on page 6, line 166, the new molecule (7b) synthesized by this method has been provided in 20% yield (Figure 4). Although the yield was less than ideal, this suggestion helped us to expand the application of the reaction.

We tried our best to improve the manuscript and made some changes in the manuscript. These changes will not influence the content and framework of the paper.

We appreciate for Editors and Reviewers warm work earnestly, and hope that the correction will meet with approval.

Once again, thank you very much for your comments and suggestions.

Reviewer 2 Report

In my opinion, the article was written correctly, the results presented clearly and legibly. The article is ready for publication.

Author Response

In my opinion, the article was written correctly, the results presented clearly and legibly. The article is ready for publication.

Response: Thank you very much for your trust and support of this manuscript.

We tried our best to improve the manuscript and made some changes in the manuscript. These changes will not influence the content and framework of the paper.

We appreciate for Editors and Reviewers warm work earnestly, and hope that the correction will meet with approval.

Once again, thank you very much for your comments and suggestions.

Reviewer 3 Report

Novel synthesis of dihydroisoxazoles by p-toluenesulfonic acid-participated 1,3-dipolar cycloaddition of dipolarophiles 3 with α-nitroketones

The authors reported p-toluenesulfonic acid mediated cycloaddition of nitroketones, delivering isoxazoline derivatives in high yields. A wide variety of substrates could be used as shown in Figures 3,4 and Table 2. The method described herein is very simple and easy to be handled and would be practical for the synthesis of 3-carbonylisoxazoline derivatives. Therefore, I recommend that the present manuscript is suitable for publication in Molecules after minor revision.

1.     Please add the detailed synthetic method of nitroketones if not commercially available.

2.     The term "catalyst" is not considered appropriate (main text and Table 1), because a stoichiometric amount of acid was used in this manuscript.

3.     Did the author use p-toluenesulfonic acid monohydride ? Please clarify.

4.     When alkyl nitroketones were used, what happen?

Author Response

The authors reported p-toluenesulfonic acid mediated cycloaddition of nitroketones, delivering isoxazoline derivatives in high yields. A wide variety of substrates could be used as shown in Figures 3,4 and Table 2. The method described herein is very simple and easy to be handled and would be practical for the synthesis of 3-carbonylisoxazoline derivatives. Therefore, I recommend that the present manuscript is suitable for publication in Molecules after minor revision.

  1. Please add the detailed synthetic method of nitroketones if not commercially available.

Response: First of all, thank you for your valuable comments on this manuscript. The detailed synthesis of nitroketones has been added in the General procedure for the α-Nitroketones section of the Supporting Information, on page S1.

  1. The term "catalyst" is not considered appropriate (main text and Table 1), because a stoichiometric amount of acid was used in this manuscript.

Response: We are very sorry for this problem due to the negligence of the author; thank you very much for pointing out this problem. The text and Table 1 have been modified.

  1. Did the author use p-toluenesulfonic acid monohydride ? Please clarify.

Response: Thank you for your valuable and thoughtful comments. The anhydrous p-toluenesulfonic acid used in this communication is noted on page 3, line 61.

  1. When alkyl nitroketones were used, what happen?

Response: We appreciate your valuable comments on this manuscript. In this paper, on page 6, line 166, the addition reaction of non-phenyl nitroketone with dipolarophiles has been added. The reaction can proceed normally, but the yields are slightly lower, 23% and 22%, respectively, as shown in Figure 4. Although the yields were less than ideal, this suggestion helped us to expand the application of the reaction. Sincere thanks again for your valuable comments.

We tried our best to improve the manuscript and made some changes in the manuscript. These changes will not influence the content and framework of the paper.

We appreciate for Editors and Reviewers warm work earnestly, and hope that the correction will meet with approval.

Once again, thank you very much for your comments and suggestions.

Round 2

Reviewer 1 Report

The manuscript requires minor change i.e., Figure 03 should be Scheme 01 and Figures 4 & 5 should be Scheme 02 & 03 respectively. Moreover, in supplementary files, the structures of compounds should come below the name of compound not above.

Author Response

The manuscript requires minor change i.e., Figure 03 should be Scheme 01 and Figures 4 & 5 should be Scheme 02 & 03 respectively. Moreover, in supplementary files, the structures of compounds should come below the name of compound not above.

Response: First of all, thank you for your valuable suggestions! The authors have changed Figure 3, Figure 4, Figure 5, and Figure 6, respectively, to Scheme 1, Scheme 2, Scheme 3 and Scheme 4. In the supplementary files, the name of compounds and the structure position of compounds have also been modified. And once again, thank you for your guidance.

We tried our best to improve the manuscript and made some changes in the manuscript. These changes will not influence the content and framework of the paper.

We appreciate for Editors and Reviewers warm work earnestly, and hope that the correction will meet with approval.

Once again, thank you very much for your comments and suggestions.